# Therapeutic Potential of Protein Tyrosine Kinase 6 in Colorectal Cancer

**DOI:** 10.3390/cancers15143703

**Published:** 2023-07-21

**Authors:** Samanta Jerin, Amanda J. Harvey, Annabelle Lewis

**Affiliations:** 1Centre for Genome Engineering and Maintenance, Division of Biosciences, Department of Life Sciences, College of Health and Life Sciences, Brunel University London, Uxbridge UB8 3PH, UK; 2138092@brunel.ac.uk; 2Centre for Genome Engineering and Maintenance, Institute for Health Medicine and Environments, Brunel University London, Uxbridge UB8 3PH, UK; amanda.harvey@brunel.ac.uk

**Keywords:** PTK6, colorectal cancer, tyrosine kinase inhibitors

## Abstract

**Simple Summary:**

Protein tyrosine kinase 6 (PTK6) is a biomarker of poor prognosis in breast cancer, but its importance in other cancer types is unclear. In this review, we explore a potential role for PTK6 in colorectal cancer. PTK6 phosphorylates multiple substrates, including signal transduction proteins, RNA binding proteins, and transcription factors, many of which are involved in key colorectal cancer pathways. PTK6 is overexpressed in colorectal tumors vs. normal tissue and activates WNT signaling when localized to the membrane in colorectal cancer cells. It is also upregulated and promotes cell survival under DNA-damaging conditions. We propose that targeting PTK6 therapeutically in colorectal cancer could increase cancer cell sensitivity to standard chemotherapy treatment.

**Abstract:**

PTK6, a non-receptor tyrosine kinase, modulates the pathogenesis of breast and prostate cancers and is recognized as a biomarker of breast cancer prognosis. There are over 30 known substrates of PTK6, including signal transducers, transcription factors, and RNA-binding proteins. Many of these substrates are known drivers of other cancer types, such as colorectal cancer. Colon and rectal tumors also express higher levels of PTK6 than the normal intestine suggesting a potential role in tumorigenesis. However, the importance of PTK6 in colorectal cancer remains unclear. PTK6 inhibitors such as XMU-MP-2 and Tilfrinib have demonstrated potency and selectivity in breast cancer cells when used in combination with chemotherapy, indicating the potential for PTK6 targeted therapy in cancer. However, most of these inhibitors are yet to be tested in other cancer types. Here, we discuss the current understanding of the function of PTK6 in normal intestinal cells compared with colorectal cancer cells. We review existing PTK6 targeting therapeutics and explore the possibility of PTK6 inhibitory therapy for colorectal cancer.

## 1. Introduction

Colorectal cancer (CRC) is the third leading cancer worldwide, with an estimated 1.9 million incidence cases and 0.9 million global deaths in 2020 [1]. The burden is expected to reach approximately 3.3 million new cases in 2040 [2]. CRC is primarily initiated in the mucosal lining of the colon and rectum by a series of genetic alterations affecting key signaling pathways, including WNT, MAPK, and BMP, accompanied by defective DNA repair pathways or cell cycle checkpoints [3,4,5]. Prognosis and treatment response are good for early stages but remain poor for late-stage metastatic cancers.

Protein tyrosine kinases are central signaling components of many pathways involved in CRC and transmit signals through the phosphorylation of tyrosine residues in target proteins. Aberrant protein tyrosine kinase signalling has frequently been linked with CRC pathogenesis [6,7]. There are two sub-families of protein tyrosine kinases (PTKs): receptor tyrosine kinases (RTKs) and non-receptor tyrosine kinases (NRTKs) [8]. RTKs activate numerous signaling pathways within cells and act both as transmembrane receptors and catalytic enzymes that contribute to regulating key cellular characteristics [9]. Examples of critical RTKs in CRC include EGFR, VEGFR, PDGFR, IGFR, and FGFRs, with both specific RTK inhibitors and multi-kinase inhibitors used to target them in patients with metastatic CRC (reviewed in [10]).

NRTKs are intracellular proteins that can be membrane-bound or nuclear-specific [7]. NRTKs are an integral component of most signal transduction cascades, regulating important cellular processes, including survival, cell division, regulation of gene expression, suppression of cell development, and regulation of cell adhesion and proliferation [8]. The functions of NRTKs are highly dependent on intracellular localization and signaling by cell surface receptors and immune system receptors [11]. NRTKS, such as Src family kinases, are already known to play a critical role in CRC [12] along with FAK [13] and JAK [14] families. However, few specific NRTK inhibitors have been approved for CRC treatment to date.

### 1.1. Protein Tyrosine Kinase 6

Protein tyrosine kinase 6 (PTK6), also known as breast tumor kinase (BRK), is an intracellular tyrosine kinase containing SRC homology domains and high sequence similarity to the SRC family. However, due to characteristic exon-intron structures shared by the PTK6, SRMS, and FRK proteins but different from those of the SRC family genes, it has been concluded that these proteins belong to a separate, BRK, tyrosine kinase family, distinct from the SRC family of kinases [15,16,17,18,19]. The full-length PTK6 isoform was originally detected in melanocytes, and the sequence was cloned from metastatic breast tumors [20,21]. Around the same time, the same isoform (alternatively named Sik) was also cloned from the intestine [22,23]. Since then, PTK6 has been detected in several normal and tumor tissues at varying levels, where it activates a variety of oncoproteins that promote cell growth, survival, and malignant transformation [21,24]. PTK6 phosphorylates both nuclear and cytoplasmic substrates such as β-catenin, PSF, Sam68, AKT, STATs, Paxillin, and IRS4 as a phosphorylating kinase, which links PTK6 to signaling pathways that serve as direct regulators of gene expression [25,26,27,28,29,30,31,32,33].

### 1.2. PTK6 Transcript Variants

The human PTK6 gene lies on chromosome 20q13.33 and consists of 8 exons that span ten kilobases with a predicted molecular weight of 50kDa [34,35] (Figure 1). It encodes approximately 451 amino acids and contains a unique kinase domain, joined by a linker region, as well as two Src homology (SH) domains: phospho-tyrosine binding and protein–protein interaction domain SH2, linker interacting and substrate recognition domain SH3. Both SH2 and SH3 domains contribute to the autoinhibition of PTK6 and protein–protein interactions. However, the primary role of SH2 is likely to be the regulation of the catalytic activity of PTK6 [36]. The SH3 domain primarily functions as a substrate recognition site but also contributes to enzyme control [37,38]. In response to changes in pH, the PTK6 SH3 domain undergoes conformational changes, indicating that its structure may impact substrate and protein interactions [36,38]. The tyrosine kinase domain of PTK6 contains the ATP binding pocket, including the key lysine residue 219, which is required for ATP-binding and phosphotransfer. Two further important residues can undergo phosphorylation and are important for activation status: tyrosine 342 (Y342) and tyrosine 447 (Y447). pY342 functions to increase PTK’s activity, while pY447 suppresses it [38,39]. Interestingly, unlike the Src family proteins, the N-terminal of PTK6 does not contain a myristoylation signal sequence and is, therefore, not specifically targeted to the membrane [16]. 

The shorter isoform of PTK6 is called the alternatively spliced variant of PTK6 (ALT-PTK6). ALT-PTK6, also known as λm5, encodes a 134-amino-acid protein [34]. It is 15 kilodaltons and shares the first 77 amino acids with the full-length version but lacks exon 2, resulting in an early stop codon. ALT-PTK6 is missing the SH2 domain and contains a unique proline-rich carboxy-terminal sequence that differs structurally from the full-length PTK6 [34]. It is reported that both PTK6 and ALT-PTK6 share a functioning substrate-recognition SH3 domain, making them competitive SH3 binding partner inhibitors [34,40]. Therefore, it is possible that ALT-PTK6 could act as a dominant negative version of PTK6 when they are co-expressed. An example of this effect has been shown in the AMPK/p27Kip1 pathway, where ALT-PTK6 blocks Y88 phosphorylation of p27Kip1, resulting in the inhibition of both CDK4 and CDK2 activity and cell cycle arrest [41]. Co-expression and altered localization of ALT-PTK6 and full-length PTK6 have been detected in human colon cancer cell lines [40].

## 2. Regulation of PTK6 Function

The function of PTK6 differs significantly depending on a number of factors: the expression level, intracellular location, phosphorylation status of PTK6 tyrosine residues, kinase activity, and the interactions with alternative splice forms [36,37,38,40,42].

### 2.1. PTK6 Expression

PTK6 expression has been identified in normal intestinal epithelial cells, skin, the oral cavity, and the normal prostate and at a low or undetectable level in normal mammary glands, lung, and lymphoma cells [20,21,23,24,43,44]. PTK6 expression is highest in the epithelia of the intestine and skin [45]. Recently, it has been discovered that PTK6 is highly expressed in the non-tumorous larynx and esophagus tissues [46,47]. Elevated PTK6 expression has been detected in multiple cancers such as breast, prostate, cervical, non-small cell lung, high-grade ovarian, bladder, pancreatic, and colon cancer, where it can promote malignant transformation and progression [34,39,40,48,49,50,51].

Figure 2 shows the expression profile of PTK6 across normal and cancerous tissues from the breast, prostate, colon, and rectum using data from the Cancer Genome Atlas and the Genotype-tissue expression portal (GEPIA database—http://gepia.cancer-pku.cn/ (accessed on 18 May 2023) [52]). Significant differences in expression are evident in breast cancer, with PTK6 expressed more highly in tumors than normal tissue as expected, but not in prostate cancer despite evidence suggesting the importance of PTK6. Of relevance to this review, significant differences are observed in colon and rectal cancers, although there is much greater variability in the normal tissue.

Figure 3 shows that PTK6 full length and ALT-PTK6 expression is variable across different colorectal cell lines indicating that its importance could be restricted to a subset of colorectal cancers.

### 2.2. PTK6 Localization

Recent research indicates that the functions of PTK6 are context-dependent and vary by cell type, tumor status, and intracellular location [32,53]. Localization of PTK6 isoforms is likely to play a key role in its function, as the range of available substrates will differ between the nucleus and cytoplasm [54]. Due to the lack of myristoylation/ palmitoylation sites or a nuclear localization sequence (NLS), the localization of PTK6 varies widely depending on tissue type and external stimulation. The localization pattern also differs between normal and tumor cells [32,53,54].

Derry et al. proposed that PTK6 is primarily a cytoplasmic kinase with the ability to re-localize to the nucleus under certain conditions, following their observations in prostate tumor cells [24]. A functional distinction between membrane and nuclear location has been reported in several studies. For instance, PTK6 is found to be expressed at the membrane in human breast, prostate, liver, and lung cancer tissues [31,39]. It is also detected in the cytoplasm and nucleus of normal colonic and prostate epithelia, respectively [45,55]. In addition, overexpression of PTK6 in prostate cancer cells causes re-localization to the nucleus from the cytoplasm [24]. PTK6’s oncogenic effects are largely mediated by its intracellular localization. For example, the membrane-targeted PTK6 enhances cell proliferation, tumorigenicity, and invasiveness in breast, prostate, HEK239 human kidney, and SW620 CRC cell lines [24,28,29,54,55]. At the cell membrane, PTK6 interacts with regulators of growth and migration such as the ErbB family transmembrane receptors, ADAM 15 A/B, and IGF-1R receptors, as well as membrane translocated cytoplasmic proteins such as paxillin, Akt, FAK, p130Cas and IRS4 (see Table 1). On the other hand, nuclear PTK6 phosphorylation causes a tumor suppressive role. One potential rationale for this role is that PTK6 nuclear targets retain the kinase in the nucleus, preventing it from enhancing growth, and exerting a tumor-suppressive action.

### 2.3. PTK6 and Substrate Phosphorylation

PTK6 has a variety of direct targets, which it phosphorylates at specific tyrosine residues (Table 1). Qiu et al. found that the PTK SH3 domain interactions are likely to govern the specificity of PTK6 substrate phosphorylation [38]. To date, more than 30 potential PTK6 substrates have been identified (Table 1). These include the RNA-binding proteins Sam68, SLM-1, SLM-2, and PSF, transcription factors STAT3 and STAT5a/b, and a variety of signaling molecules EGFR, p190RhoGAP, paxillin, Akt, IRS-4, BKS/STAP-2, and KAP3A.

### 2.4. PTK6 and Signalling Molecule Interaction

PTK6 functions downstream of ERBB2 (HER2) and other receptor tyrosine kinases, such as EGFR and MET [62,76,77,78]. Since ERBB2 forms heterodimers with EGFR, and MET can heterodimerize with both ERBB2 and EGFR [79], it is not clear if MET and EGFR activate PTK6 directly or act through ERBB2. However, activated PTK6 was found to directly phosphorylate tyrosine 845 in the EGFR kinase domain, suggesting a feedback loop involving the two proteins [61]. PTK6 has also been found to bind to and phosphorylate the tyrosine residue on ARAP1. PTK6 enhances EGFR signaling via ARAP1 by inhibiting EGFR internalization and degradation [57,59,61]. Clinically, the interaction between PTK6 and EGFR has wide implications, as PTK6 may potentially be a factor in the low efficacy of anti-EGFR drugs in breast cancer treatment. Indeed, depletion of PTK6 was found to sensitize cells to Cetuximab [61]. Furthermore, EGFR is overexpressed in CRC, and anti-EGFR therapies are given to patients with metastatic CRC [80]. Therefore, PTK6 could also be explored as a predictive biomarker for the efficacy of EGFR inhibitors in CRC. Serine/threonine kinase AKT is another direct substrate of PTK6, although studies conflict as to whether it has a repressive or activating role [31,62,81,82,83]. Again, AKT is a key pathway and potential therapeutic target in CRC [84]. PTK6 directly phosphorylates a number of other key signal transduction proteins, including paxillin which activates Rac1 GTPase [25] and insulin receptor substrate 4 (IRS-4) downstream of IGF-1 [30].

### 2.5. PTK6 and RNA Binding Protein Interaction

PTK6 binds and phosphorylates several nuclear RNA-binding proteins, including Sam68, which was one of the first PTK6 substrates identified and the most extensively studied [26]. PTK6 expression negatively regulates Sam68 by direct phosphorylation of specific tyrosine residues [26]. Sam68 is important in CRC due to its regulation of the DNA damage response [85]. Similar to Sam68, the Sam68-like mammalian proteins SLM-1 and SLM-2 are phosphorylated by PTK6 in vitro, negatively regulating their RNA-binding function [45]. PTK6 was shown to phosphorylate and yield a cytoplasmic re-localization of the PTB-associated splicing factor (PSF) from the nucleus, resulting in cell cycle arrest, a function consistent with reports of the pro-tumor suppressive role of PTK6 within the nucleus [28].

### 2.6. PTK6 and Transcription Factor Interaction

PTK6 has been involved in activating the Signal Transducer and Activator of Transcription (STAT) family of proteins [27,32]. STAT3 activation has been associated with a poor prognosis in breast cancer due to higher cell proliferation, migration, and survival [27,86,87,88]. PTK6 activates STAT3-mediated transcription [32,89,90] and may also activate STAT5-mediated transcription [91]. Both transcription factors are directly phosphorylated by PTK6 and STAT3, when activated, promotes proliferation and migration and impairs apoptosis in colon cancer cell lines, mouse models, and breast cancer cell lines MDA-MB-231 and T47D [27,32].

### 2.7. Kinase Activity of PTK6

In addition to localization, the oncogenic functions of PTK6 depend on the phosphorylation status of its two tyrosine residues. Phosphorylation of the Y342 residue activates PTK6, with active PTK6 being detected in breast tumors but not in normal breast tissues [39]. Conversely, in ovarian cancer, PTK6 de-phosphorylation of Y342 occurs due to tyrosine phosphatase 1B (PTP1B) activity [92]. Similarly, in vitro studies on prostate cancer demonstrate that PTK6 is also dephosphorylated at Y342 by the phosphatase and tensin homolog (PTEN) [73]. On the other hand, Y447 phosphorylation inhibits kinase activity [38,42], suggesting that specific tyrosine kinase residue phosphorylation states could play a vital but tissue-specific, role in cancer development and progression. Further regulation of kinase activity occurs in the linker region via conserved trytophan W184 [38]. PTK6 autophosphorylation and kinase activity have been shown to be activated by EGF, heregulin-β1, and IGF1 [69,93,94]. The kinase activity of PTK6 is negatively regulated by SRMS kinase [95], as well as the STAT3 target SOCS3 [96].

In several instances, activities of PTK6 have been shown to be kinase-independent and may be attributed to adaptor/scaffolding functions mediated by its SH3 and SH2 domains. Indeed, mouse xenografts of breast cancer cells carrying a kinase-inactive PTK6 (due to K219M mutation) still developed into tumors at a similar rate to those with wildtype PTK6 [97]. Kinase-inactive PTK6 was also shown to promote the proliferation of the T47D breast cancer cell line. PTK6 kinase activity was not required for PTK6-dependent HGF-induced cell migration or PTK6-ERK5 interaction and ERK5 activation [76]. Similarly, PTK6 was able to promote epithelial characteristics in colon cancer cell lines independent of kinase activity [42]. These data suggest that PTK6 has both kinase-dependent and independent functions in normal and cancer tissue. Some kinase-independent functions of PTK6 may play a role in oncogenesis, and specific inhibitors may not yield any anticancer efficacy [98].

Importantly though, two studies have shown that in triple-negative breast cancer cell lines, the inhibition of PTK6 kinase activity augmented the cytotoxic effects of both doxorubicin and paclitaxel. Therefore, it is likely that kinase inhibition of PTK6, in combination with chemotherapies, would still be effective therapeutically but may depend on tumor context [42,97].

### 2.8. Isoform Interaction

It is reported that both PTK6 and ALT-PTK6 share a functioning SH3 domain, making them competitive SH3 binding partner inhibitors [34,40]. This functional relationship is likely to be critical for the functional roles of PTK6 and ALT-PTK6 in cells when they are co-expressed. Co-expression and altered localization of ALT-PTK6 and full-length PTK6 have been detected in human colon and prostate cancer cell lines [40]. Although the role of ALT-PTK6 has been broadly explored in prostate cancer cells, its expression pattern, involvement in regulating signaling, and mechanisms of action in CRCs have yet to be determined.

In co-transfection experiments with full-length PTK6 and ALT-PTK6, Brauer et al. demonstrated the inhibitory effects of ALT-PTK6 on the phosphorylation of PTK6 in prostate cancer cell lines [40]. They suggest that ALT-PTK6 inhibits the transcriptional activities of TCF/β-catenin, thereby inhibiting cell growth and proliferation. In addition, increasing levels of ALT-PTK6 are associated with decreased phosphorylation of PTK6 and enhanced nuclear function of PTK6 in prostate cancer [40]. This study revealed that ALT-PTK6 has a functional role in PTK6 signalling in prostate cancer by negatively regulating PTK6 activity and subcellular localization. This information implies that ALT-PTK6 inhibits the full-length form of PTK6 as well as its cytoplasmic and membrane-associated substrate-binding ability [40]. Recently the ratio of ALT-PTK6 to full-length PTK6 expression was shown to have significant prognostic value in predicting patient outcomes in breast cancer [97]. Therefore, further detailed investigation of ALT-PTK6 and PTK6 interactions in cancers may be a fruitful avenue of research to develop a highly-specific PTK6 inhibitor.

## 3. PTK6 Expression and Activation in Colon Cancer

### 3.1. Normal Intestine

PTK6 is present throughout the normal human gastrointestinal tract, in the esophagus, stomach, duodenum, and colon. In neonatal mice, the highest levels of expression are in the colon, but in the adult mouse highest expression is in the ileum of the small intestine. In both regions, expression is concentrated in the more differentiated non-dividing epithelial cells [43,99]. Disruption of PTK6 by germline knockout suggests that, in the normal mouse small intestine, PTK6 acts to promote growth arrest and maturation of columnar epithelial cells [43]. In fact, PTK6 directly regulates β-catenin transcriptional activity, repressing the WNT pathway to promote enterocyte differentiation in the mouse intestine [43]. PTK6 may also play a role in driving differentiation in cancer cells since increased PTK6 expression is also evident in differentiating Caco-2 colon cancer cells [99].

### 3.2. PTK6 in Colon Cancer

PTK6 has been implicated in modulating cell signaling in numerous tissues. However, it is becoming clear that there are functional distinctions between its role in normal and tumor cell types. In normal epithelial tissues, PTK6 expression supports cell differentiation and tissue homeostasis reviewed in [55]. However, in cancers such as breast and ovarian, PTK6 is involved in cellular proliferation, migration, and survival activities [48,49,69].

Nevertheless, there is still some uncertainty surrounding the role of PTK6 in colon cancer. Llor et al. demonstrated that PTK6 mRNA expression is low in normal tissues compared to the adjacent tumor tissues at different stages, with tumor tissues showing 2–3.5 times higher PTK6 expression than the normal tissues [99]. In addition, it is reported that the PTK6 gene is amplified in colon cancer [42]. Conversely, Palka-Hamblin et al. [29] showed that nuclear-targeted PTK6 mediated phosphorylation of β-catenin and inhibited β-catenin/TCF-driven transcription and WNT pathway activity in SW620 colon cancer cells. However, membrane-targeted PTK6 positively regulated β-catenin/TCF transcriptional activity suggesting that membrane localization enhances PTK6 oncogenic activity [29]. Similar conclusions were drawn by Mathur et al. [42], who reported decreased PTK6 expression in colon carcinoma samples relative to normal differentiated epithelial cells but elevated PTK6 in the membrane of the metastatic colon cancer cells. However, Mathur et al. showed that PTK6 also played a role in maintaining epithelial characteristics of colon cancer cells, similar to normal intestinal cells; knockdown of PTK6 in SW480 cells led to EMT and increased xenograft tumor growth. The same study reported that TCGA Colorectal Cancer dataset analysis, comparing tumor with paired normal tissue, clearly demonstrated PTK6 overexpression in colon adenocarcinomas relative to adjacent normal tissue [42], which is similar to the unpaired TCGA/GTex comparison (Figure 1). It must be noted that the studies above combine analysis at both the mRNA and protein levels, which may contribute to some of the conflicting observations. However, it is clear that the importance of PTK6 activation status and localization are also key factors in mediating its pro or anti-tumorigenic effects.

In vivo, experiments by Haegebarth A et al. suggested that disruption of the PTK6 gene has no effect on intestinal tumorigenesis in mouse models [43]. Interestingly, also in mouse models, loss of PTK6 conferred resistance to tumor development following treatment with the colon carcinogen azomethane (AOM) [32]. Similarly, when DNA-damaging agents such as radiation and chemotherapy drugs (e.g., 5-fluorouracil) were used in colon cancer cells, PTK6 expression promoted survival of these cells, again suggesting an important role for PTK6 following treatment with DNA damaging agents [100]. Furthermore, it was found that γ-Irradiation induces PTK6 expression in the intestinal crypts of proliferating progenitor cells, where it is implicated in DNA-damage-induced apoptosis [53]. Thus, PTK6 expression appears to be induced in response to DNA damage and, in some cases, promotes survival, but in others, apoptosis.

PTK6 actively regulates cell cycle properties in the breast and colon via activating the STAT (Signal Transducer and Activator of Transcription) family proteins [27,100]. Following activation of PTK6 by DNA damage, PTK6 stimulated STAT3 in the HCT116 colon cancer cell line [100]. In addition, the disruption of PTK6 in mice impaired STAT3 activation and protected animals from AOM/DSS-induced colon cancer, indicating that this substrate could be a key mediator for PTK6’s role in DNA damage-induced CRC [32].

The above data indicate a complex and context-dependent mode of PTK6 function in CRC. Additional research is required to clarify the precise functions of PTK6 in normal intestine and colon tumors, including levels of expression, cellular localization, and response to cellular and DNA damaging stresses.

## 4. Tyrosine Kinase Targeted Therapy in Cancer

Since aberrant tyrosine kinase activities are directly linked with poor clinical outcomes and survival rates for cancer patients, they have become one of the most extensive classes of therapeutic targets under development. Already, there is a range of clinically approved therapeutic tyrosine kinase inhibitors available, usually used in combination with chemotherapy, radiotherapy, and surgery (http://www.brimr.org/PKI/PKIs.htm (accessed on 16 May 2023)). Plenty of others are in the drug discovery and clinical trials pipeline.

There are two main types of protein tyrosine kinase inhibitors, reviewed in [101]. Type 1 inhibitors competitively target the ATP binding pocket in active kinases. However, they often exhibit low selectivity, which increases the potential for adverse side effects. Type 2 inhibitors are more selective and target the inactive conformation of kinases to stabilize the enzyme in the inactive state. More recently, allosteric and covalent-type kinase inhibitors have shown even greater target selectivity [102,103]. 

Understanding the critical role of tyrosine kinases in the development of CRC has created a platform to identify, characterize and test a plethora of therapeutic kinases inhibitor molecules. Of these, monoclonal antibody inhibitors of EGFR (e.g., Cetuximab) and VEGFR (e.g., Bevacizumab) are the most widely used globally, while the multi-kinase inhibitor regorafenib is also FDA-approved [10]

### 4.1. PTK6 Inhibitors

Despite the growing evidence of PTK6’s oncogenic function in cancers, there have been few conclusive studies on the clinical benefits of targeted PTK6 inhibitors. Here we summarize the available inhibitors and current progress toward clinical utility (Table 2).

#### 4.1.1. Biological Inhibitors

The suppressor of cytokine signaling 3 (SOCS3) protein has been identified as the first potent biological inhibitor of PTK6, showing a tumor suppressive effect in T47D and MDA-MB-231 breast cancer cell lines [96]. Gao et al. show that inhibition of PTK6 activity by SOCS3 is mediated by binding to the kinase domain, causing PTK6 ubiquitination and protein degradation. Since PTK6 activates STAT3 and STAT3 induces SOCS3 expression, this results in a negative feedback mechanism to control PTK6 activity [96]. A very recent study on uveal melanoma cells demonstrates that overexpression of SOCS3 can partially inhibit the PTK6-driven uveal melanoma cell proliferation [104].

#### 4.1.2. Chemical Inhibitors

To date, some of the most potent PTK6 inhibitors, causing significant repression of breast cancer cell migration and invasion, are marine natural products such as the derivatives of the marine Triterpene Sipholenols: 4β-O-benzyl sipholenol A and 4β-O-benzyl-19,20-anhydrosipholenol A [105]. The compounds were identified through a semi-synthetic optimization of the triterpene sipholenols and were found to have anti-migratory and anti-invasive effects on MDA-MB-231 breast cancer cells while remaining non-toxic to the normal breast epithelial cells. These two derivatives exhibited inhibition of PTK6 phosphorylation in vitro. Oleanolic acid is another common therapeutic triterpene derived from Terminalia bentzoe L. [106]. Again, semi-synthetic optimization of oleanolic acid identified two potent analogs: 3-O-[N-(3′-chlorobenzenesulfonyl)-carbamoyl]-oleanolic acid and 3-O-[N-(5′-fluorobenzenesulfonyl)-carbamoyl]-oleanolic acid. Significantly reduced phosphorylation of PTK6 and its substrates was observed, accompanied by reduced invasiveness and migration of breast cancer cells. Further natural marine products have been identified using similar technologies and are also proposed to exert their anti-proliferative and anti-migratory effects via PTK6 and paxillin. These include derivatives of phenylmethylene hydantoins and Z-4-hydroxyphenylmethylene hydantoin [107].

Perhaps surprisingly, Geldanamycin, a known suppressor of Heat-shock protein 90 (Hsp90), has also been shown to mediate its effects by targeting PTK6 for proteasomal degradation [68]. The inhibitor is thought to decrease Hsp90/PTK6 interactions and therefore increase PTK6 interactions with E3 ligase components of the proteasome.

Chemical inhibitors of the PTK6 ATP binding pocket show some of the highest selectivity for PTK6. Tilfrinib (4f) 4-anilino α-carbolines mediated PTK6 inhibition shows anti-proliferative effects on MCF7, HS-578/T, and BT-549 breast cancer cell lines [108]. Mahmoud et al. demonstrated that 4f significantly reduces the phosphorylation of PTK6 substrate STAT3 and induces cell death of non-adherent breast cancer cells. MK138 and MK150, the two most potent derivatives of Tilfrinib, selectively inhibit PTK6 and suppress STAT3 activity in T47D breast cancer cell lines [109]. Similarly, the compound Imidazo [1,2-a]pyrazin-8-amines is highly selective at attenuating the phosphorylation of PTK6 substrate SAM68, thereby inhibiting PTK6 activity [110]. Shim et al. [111] reported derivatives of (E)-5-(benzylideneamino)-1H-benzo[d]imidazol-2 (3H)-one) that showed 20-fold higher selectivity than similar NRTK’s. These compounds effectively decreased the phosphorylation of PTK6 substrates Paxillin and STAT3 [111]. Findings from Qiu et al. demonstrated very low selectivity of type 1 inhibitors, compounds 21a and 21c, with the type 2 inhibitors, PF-6683324 and PF-6689840, showing superior selectivity [98]. 

Small molecule inhibitor XMU-MP-2 also binds to the ATP binding site of the PTK6 and acts as a potent PTK6 inhibitor with low cytotoxicity [112]. The drug suppressed the growth of tumors induced by PTK6-transformed cells in xenograft models and suppressed both STAT3 and STAT5 activity. XMU-MP-2 also displayed strong synergy with HER2 and ER inhibitors. This is encouraging since PTK6 inhibitors, like many tyrosine kinase inhibitors, are likely to be most effective as part of combination therapies [112].

Further attempts to design direct PTK6 inhibitors with high specificity have led to the identification of Pyrazolopyrimidine PP1 and PP2 [111] and PF-6683324 and PF-6689840, which were designed specifically to bind to unphosphorylated PTK6 [98]. However, despite growth inhibition of MDA-MB-231 breast cancer cells, the direct inhibition of PTK6 could not be demonstrated, and off-target effects were evident.

**Table 2 cancers-15-03703-t002:** Compounds that target PTK6 activity.

Name of the Inhibitor	Type of Inhibitor	References
SOCS3(The suppressor of cytokine signalling 3)	Biological	[96,104]
4β-O-benzyl sipholenol A and 4β-O-benzyl-19,20-anhydrosipholenol A	Marine natural products	[105]
Oleanolic acid	Marine natural products	[106]
Phenylmethylene hydantoins and *Z*-4-hydroxyphenylmethylene hydantoin	Marine natural products	[107]
Geldanamycin (an inhibitor of heat shock protein 90 (HSP90)	Natural product	[68]
Tilfrinib (4f)	Chemical	[108]
Imidazo [1,2-a]pyrazin-8-amines	Chemical	[110]
(E)-5-(benzylideneamino)-1H-benzo[d]imidazol-2 (3H)-one)	Chemical	[111]
XMU-MP-2	Chemical	[112]
Pyrazolopyrimidine PP1 and PP2	Chemical	[111]
PF-6683324, PF-6689840, 21a, 21c	Chemical	[98]
Dasatinib	Chemical	[113,114]
Vemurafenib	Chemical	[115]

Beyond these novel and newly synthesized compounds, two drugs already used widely in the clinic also appear to target PTK6. Dasatinib is a small molecule inhibitor of BCR-ABL and SRC family tyrosine kinases that has been widely used for the treatment of chronic myeloid leukemia [113,114]. Dasatinib has been described as an “off-target” tyrosine kinase inhibitor of PTK6, as the drug also significantly reduces PTK6 activity [116]. Unfortunately, the mode of action of Dasatinib against PTK6 has not been explored; however, it is important to note that Dasatinib is also highly effective against other kinases such as bone marrow kinase on chromosome X (BMX) and bruton tyrosine kinase (BTK), and therefore quite general off-target effects can be observed in clinical settings [117]. Vemurafenib, designed as an inhibitor of BRAFV600E, was also shown to selectively inhibit PTK6 through binding to its active site [115]. Vemurafenib (PLX4032) inhibits PTK6 in both prostate and colon cancer cells and was able to reduce tumor growth in prostate cancer xenograft models [42,115]. Vemurafenib inhibition of PTK6 presents an interesting avenue for follow-up studies in CRC models since BRAF-activating mutations frequently occur in some subtypes of CRC.

## 5. Future Perspectives: The Potential of PTK6 Inhibitory Therapy in CRC

The role of PTK6 in CRC remains context-dependent and complex. Here, we have summarized and discussed the often conflicting literature on PTK6 in colon cancer. A number of lines of evidence combine to provide a compelling argument for investigating PTK6 as a potential therapeutic target in CRC. Publicly available data suggest that PTK6 is overexpressed in tumors compared with normal tissue. When located at the membrane (but not the nucleus), PTK6 has been shown to increase WNT pathway activity, where WNT is the most frequently upregulated CRC pathway and best-characterized driver of CRC. This membrane localization is more apparent in metastatic colon cancer cells. What is particularly interesting is that PTK6 expression promotes the survival of cells in DNA-damaging conditions. This suggests that repressing PTK6 expression in combination with chemo and radiotherapies could enhance their efficacy. 

As with all tyrosine kinase inhibitors, PTK6 inhibitors exhibit a range of potencies and selectivities. One of the most selective chemical inhibitors is tilfrinib (4f). Tilfrinib’s effects are mediated through a reduction in phosphorylation and activity of the transcription factor STAT3. STAT3 is overexpressed in about 70% of human cancers and has been shown to be important for the survival of CRC stem-cell-like cells [118].

CRC cancer cell lines show a wide range of PTK6 expression levels. In our laboratory, we have begun experiments to measure the effects of PTK6 inhibitors, including tilfrinib, on CRC cell line proliferation and survival. Preliminary evidence suggests that PTK6 inhibitors are cytotoxic to cell lines with high PTK6 levels when used alone and also act in synergy with chemotherapy drugs such as 5-FU. Further experiments are underway to explore the levels and localization of PTK6 and ATL-PTK6 in these cells, how these relate to PTK6 inhibitor sensitivity, and whether PTK6 expression levels are modulated during treatment with chemo and radiotherapies.

## 6. Conclusions

PTK6 has long been known as a prognostic biomarker in breast cancer, but its importance in other cancers is less clear, and the available data are complex and sometimes conflicting. Despite this, we argue here that it is worth exploring PTK6 as a therapeutic target in CRC and making use of the plethora of CRC model systems to test some of the available PTK6 inhibitors.

## Figures and Tables

**Figure 1 cancers-15-03703-f001:**
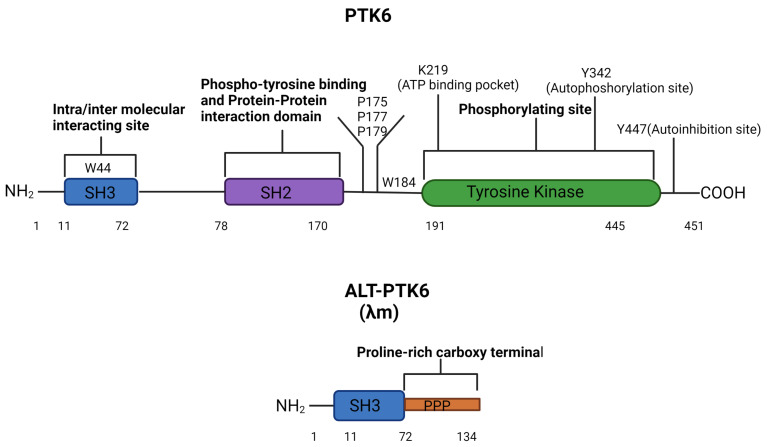
PTK6 domains structure and regulatory residues. The full-length PTK6 protein is made up of a 451 amino acids chain with an N-terminal SH3 domain, an SH2 domain, and a C-terminal tyrosine kinase domain. The tyrosine kinase domain consists of three regulatory residues K219 (in the ATP binding pocket), Y342 (autophosphorylation site), and Y447 (autoinhibition site). Mutation of K219 results in a kinase-dead PTK6, mutation of Y342 reduces the activation of PTK6, and mutation of Y447 makes a constitutively active PTK6. W44 in the SH3 domain and P175, P177, P179, and W184 in the linker are key for intra-molecular interactions affecting catalytic activity [37]. The alternative splice form of PTK6 (ALT-PTK6) is 134 amino acids long, consisting of an SH3 domain linked with a proline-rich carboxy-terminal tail.

**Figure 2 cancers-15-03703-f002:**
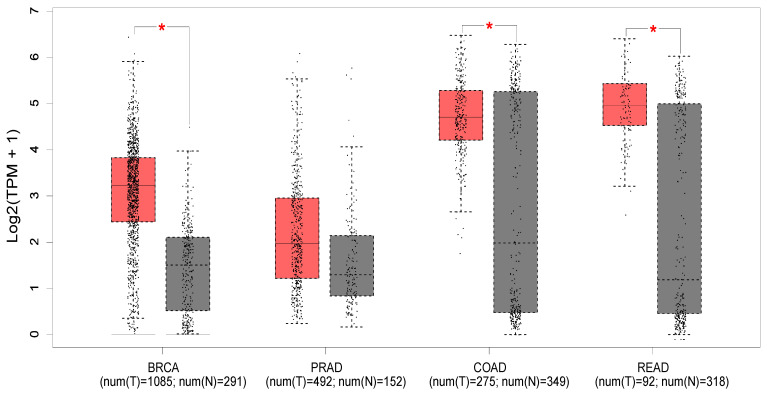
The median expression of PTK6 in TCGA malignancies (red) and GTex normal tissues (black) is displayed in a boxplot (GEPIA database—http://gepia.cancer-pku.cn/ (accessed on 18 May 2023) [52]). Breast invasive carcinoma (BRCA), colon adenocarcinoma (COAD), prostate adenocarcinoma (PRAD), rectum adenocarcinoma (READ). * indicates *p* < 0.01 (ANOVA).

**Figure 3 cancers-15-03703-f003:**
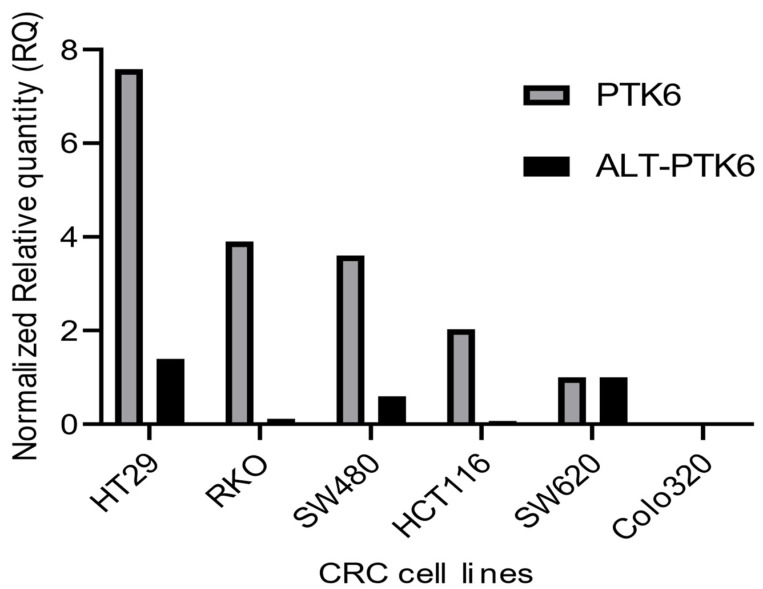
Relative mRNA expression of PTK6 transcripts in CRC cell lines. The graph depicts relative PTK6 and ALT-PTK6 expression. Data were analyzed using the ΔΔCt method and normalized to the reference gene GAPDH. The graph was plotted using GraphPad PRISM 9 program.

**Table 1 cancers-15-03703-t001:** PTK6 substrates and binding partners. •symbol represents direct substrates of PTK6; “+” or “−” indicates activation or inhibition of substrates.

Substrates and Interacting Proteins	Tissues	Localization	Promotes	References
ADAM-15A	Breast	Membrane	Unknown	[56]
ADAM-15B	Breast	Membrane	Unknown	[56]
•AKT+	Prostate/Breast	Membrane/Cytoplasm	Oncogenic	[31]
•ARAP1+	Breast	Membrane/Cytoplasm	Oncogenic	[57]
•BCAR1 (P130CAS)+	Prostate	Membrane/Cytoplasm	Oncogenic	[58]
•β-catenin+/−	Colon	Membrane/Cytoplasm Nucleus	Oncogenic Differential	[29]
c-CBL		Cytoplasm	oncogenic	[59]
Dok1	Breast	Cytoplasm/Nucleus	Oncogenic	[60]
•EGFR+	Breast	Membrane	Oncogenic	[61]
ERBB2+	Breast	Membrane	Oncogenic	[39,62]
ERBB3	Breast	Membrane	Unknown	[63]
ERBB4	Breast	Membrane	Unknown	[63]
ERK5		Cytoplasm	Cell migration	[64]
FAK+	Prostate	Membrane/Cytoplasm	Oncogenic	[65]
•FLJ39441	Breast	Cytoplasm	Unknown	[66]
GAPA p65		Cytoplasm	Differentiation	[67]
•GNAS		Cytoplasm	Cell migration	[66]
HSP70		Cytoplasm	Protein stability	[68]
HSP90		Cytoplasm	Protein stability	[68]
IGF-1R+	Breast	Membrane		[69]
•IRS-4+	Breast	Membrane/Cytoplasm	Oncogenic	[30]
•KAP3A+	Breast	Cytoplasm/Nucleus	Oncogenic	[66]
P38 MAPK+	Breast	Cytoplasm	Oncogenic	[70]
•p190RhoGAP+	Breast	Cytoplasm	Oncogenic	[71]
P27Kip1+	Breast	Cytoplasm/Nucleus	Oncogenic	[72]
•Paxillin+	Breast	Membrane/Cytoplasm	Oncogenic	[25]
PSF+	Breast	Nucleus	Cell cycle arrest	[28]
PTEN	Breast/Prostate	Cytoplasm	Unknown	[63,73]
•SAM68+/−	Breast/Colon	Nucleus	Differential	[26,28]
•SLM1+/−	Mammary gland	Nucleus	Differential	[45]
•SLM2+/−	Mammary gland	Nucleus	Differential	[45]
•SRC+	Breast/Prostate	Membrane	Oncogenic	[74]
•STAP2+	Breast	Cytoplasm	Oncogenic	[21]
STAP3+		Cytoplasm		
•STAT3	Breast/Colon	Cytoplasm/Nucleus	Oncogenic	[27]
•STAT5a/b+	Breast	Cytoplasm/Nucleus	Oncogenic	[75]

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
