# Peer review of "Therapeutic Potential of Protein Tyrosine Kinase 6 in Colorectal Cancer"

_cancers, 2023, doi:10.3390/cancers15143703_

Round 1

Reviewer 1 Report

Jerin and colleagues review the roles of PTK6 in signaling and cancer and suggest that targeting PTK6 in colon cancer may have therapeutic benefits.  This is an interesting proposal that has merit, and some published data support that PTK6 inhibition may enhance benefits of chemotherapy. However, the biology of PTK6 expression and signaling in the gut is complex, and incorrect statements about roles for PTK6 in the intestine were included in the review.

Specific Comments

Line 62:  Based on their conserved gene structure, it has been concluded that PTK6, SRMS, and FRK, which share the same exon-intron structure that differs from the gene structure of all SRC-family members, belong to a separate tyrosine kinase family, distinct from SRC family of kinases. See the following references.

Neet and Hunter, Genes to Cells (1996) 1, 147–169

Lee et al. Mol Cells, 1998 Aug 31;8(4):401-7.

Robinson et al. 2000.  Oncogene 19, 5548 - 5557.

Serfas and Tyner.  Oncol Res. 2003;13(6-10):409-19. 

D'Aniello et al., Molecular Biology and Evolution, 2008; 25(9): 1841–1854

Line 64.   In a review focused on GI cancers, it should be noted that PTK6 (called Sik) was cloned from the intestine around the same time.

Siyanova et al. Oncogene. 1994 Jul;9(7):2053-7.

Vasioukhin et al. Oncogene. 1995 Jan 19;10(2):349-57.

Line 84.  K219 is involved in ATP-binding and phosphotransfer; it is not phosphorylated for kinase activation.

Line 103:  Possible functions of ALT-PTK6 are discussed.  A key reference demonstrating that ALT-PTK6 blocks p27 Y88 phosphorylation, thereby inhibiting CDK4 activity is missing and should be included in this discussion.

Lines 115 -119:  Prostate cancer is missing from the list. Next to breast cancer, PTK6 has been most studied in prostate cancer.

Re:  Table 1.

The reference identifying FAK as a direct PTK6 substrate is incorrect.  It should be:  Zheng et al., Oncogene. 2013 Sep 5;32(36):4304-12.

Note that p130Cas (official name BCAR1) is listed twice in the table.  The reference identifying p130Cas as a direct PTK6 substrate is Zheng et al., J Biol Chem. 2012 Jan 2;287(1):148-158. The reference cited is incorrect.

In ref 56 PTEN was shown to IP with PTK6, but it was not demonstrated to be a direct substrate of PTK6. 

It is not mentioned that SRC is a PTK6 substrate (Alwanian et al., 2022).  PTK6 was shown to phosphorylate SRC on its activation site. This may have implications for colon cancer.

Line 283. The statement is incorrect.  In fact, disruption of Ptk6 led to increased growth and delayed differentiation.  During normal homeostasis PTK6 is restricted to nondividing differentiated cells.  Germline Ptk6 knockout led to increased growth, delayed differentiation, increased beta-catenin activity (Haegebarth et al., 2006).  PTK6 also played a role in maintaining epithelial characteristics of colon cancer cells; knockdown of PTK6 in SW480 cells led to an EMT and increased xenograft tumor growth (Mathur et al, 2016).  However, after DNA damage, PTK6 expression is induced in crypt/progenitor cells where it promoted STAT3 activation and subsequent proliferation of cells with mutations leading to enhanced tumorigenesis in vivo (Gierut et al., 2011).

Lines 330-331:  The statement is incorrect. Disruption of Ptk6 in mice impaired STAT3 activation, and protected animals from AOM/DSS induced colon cancer.

Reviewer 2 Report

Jerin et al. review the current status of PTK6 as a potential therapeutic target in colorectal cancer.  Although, as they note there is considerable conflicting data and no firm conclusions about their clinical utility, the review appears to be timely and interesting. I indicate some typos and suggest a few wording changes below:

82-84 “The tyrosine kinase domain of the PTK6 contains the ATP binding pocket, containing the key lysine residue 219, which needs to be phosphorylated to activate the kinase activity.” Maybe substitute “including” for “containing” to avoid repeating contain.

125-127 “Of relevance to this review, significant differences are observed in colon and rectal cancers although there is much greater variability in the normal tissue suggesting.” Remove suggesting or complete sentence.

149-151 “Derry et al. [18] proposed that PTK6 is primarily a cytoplasmic kinase with the ability to relocalize to the nucleus under certain conditions, following their observations in prostrate tumor cells [18].” Don’t need to cite ref 18 twice in the same sentence.

190 “EGFR inhibitors in CFC” Probably should be CRC.

263-266 “They suggest that ALT-PTK6 inhibits the transcriptional activities of TCF/β-catenin, thereby inhibiting cell growth and proliferation. In addition, increasing levels of ALT-PTK6 are associated with decreased phosphorylation of PTK6 and enhanced nuclear localization in prostate cancer [34].” Enhanced nuclearization of β-catenin, PTK6 or ALT-PTK6?

282-283 “The highest level of PTK6 mRNA and protein is found in the middle and upper colonic crypts”. Do they mean upper cells of colonic crypts?

300-302 “Conversely, Palka-Hamblin et al [23] showed that nuclear targeted PTK6 mediated phosphorylation of β-catenin inhibited β-catenin/TCF driven transcription and WNT pathway activity in SW620 colon cancer cells. Should “inhibited” instead be “inhibiting”?

310-312 “Cancer dataset analysis, comparing tumor with paired normal tissue, clearly demonstrated PTK6 overexpression in colon adenocarcinomas relative to adjacent normal tissue. [35] which is similar to the unpaired TCGA/GTex comparison (Figure 1). Comma instead of a period.

Correction of a few typos needed.

Round 2

Reviewer 1 Report

There are still several errors in the manuscript. Greater attention to detail is needed from the authors. This is important for a review article, which readers will use as an important resource.  The references should be carefully checked.

Fig. 1 and Fig. 1 legend. 

Labeled residues are not discussed in the figure legend (W44, W184 P175, P177, P179). What is their significance?

The SH2 domain is not the enzyme catalysis site, the catalytic domain does the work.

Mutation of Y342 does not kill kinase activity but reduces its activation.  The K219 M mutant is kinase-dead.

The alternative splice form of PTK6 (ALT-PTK6) is NOT 1354 amino acids long. ALT-PTK6 is only a 15 kDa protein.

Lines 77 -79:  The protein description is confusing. N-terminal lipid modification does not occur in the linker. 

Lines 87 and 88:  This has not been corrected.  The lysine residue at position 219 (K219) is not phosphorylated.  It is required for ATP binding.  Mutation of K219 to M leads to kinase-dead PTK6.

Line 288:  In neonatal mouse highest levels of expression are in the colon, but in the adult mouse highest expression is in the ileum of the small intestine. In both regions, expression is concentrated in more differentiated cells under normal conditions (see Haegebarth et al., 2006, Fig. 2).

Lines 229 and 237:  These are redundant.  PTP1b and PTPN1 are the same protein. PTPN1 is the gene name.

Line 447:  Vemurafenib (PLX4032) was also used to inhibit PTK6 activity in colon cancer cells (Mathur et al., 2016, Fig. 7b).

Table 1

The heading for column one should read Substrates and Interacting Proteins.  Not all of these proteins are substrates;  co-IP of a protein with PTK6 does not indicate that it is a substrate of PTK6. 

BCAR1/p130Cas is a direct substrate

EGFR is a direct substrate.

Correct Ref.

X Li, Y Lu, K Liang, J-M Hsu, C Albarracin, GB Mills, M-C Hung and Z Fan.  Brk/PTK6 sustains activated EGFR signaling through inhibiting EGFR degradation and transactivating EGFR.   Oncogene (2012).

ERK5 correct refs

Ref. Ostrander et al., Cancer Res 2007; 67: (9). May 1, 2007, and Castro et al., 2010 (ref. 73), not ref 63 as indicated.

P38 ref is incorrect for interaction.

Prostate is misspelled as Prostrate in several places.

PTEN interaction was also demonstrated in prostate cells (Wozniak et al., 2017)

Author Response

We thank reviewer 1 for their first report and have previously made all their suggested improvements. Since we received no further comments on our revised manuscript, we are assuming that this reviewer is now satisfied that the manuscript is of an acceptable standard for publication.

Round 3

Reviewer 1 Report

Most of my previous comments have been addressed, but there are several things that still need to be corrected, and this review may not comprehensively list all errors.  Greater attention to detail is needed from the authors. This is important for a review article, which readers will use as an important resource.  The references should be carefully checked.

Fig. 1 and Fig. 1 legend. 

Labeled residues are not discussed in the figure legend (W44, W184 P175, P177, P179). What is the significance?

The SH2 domain is not the enzyme catalysis site, the catalytic domain does the work.

Mutation of Y342 does not kill kinase activity but reduces its activation. It is the K219 M mutant.

The alternative splice form of PTK6 (ALT-PTK6) is NOT 1354 amino acids long. It is only a 15 kDa protein.

Lines 77 -79:  The protein description is confusing. N-terminal lipid modification does not occur in the linker. 

Lines 87 and 88:  This has not been corrected.  The lysine residue at position 219 (K219) is not phosphorylated.  It is required for ATP binding.  Mutation of K219 to M leads to kinase-dead PTK6.

Line 288:  In neonatal mouse highest levels of expression are in the colon, but in the adult mouse highest expression is in the ileum of the small intestine. In both regions, expression is concentrated in more differentiated cells under normal conditions (see Haegebarth et al., 2006, Fig. 2).

Lines 229 and 237:  These are redundant.  PTP1b and PTPN1 are the same protein. PTPN1 is the gene name.

Line 447:  Vemurafenib (PLX4032) was also used to inhibit PTK6 activity in colon cancer cells (Mathur et al., 2016, Fig. 7b).

Table 1

The heading for column one should read Substrates and Interacting Proteins.  Not all of these proteins are substrates;  co-IP of a protein with PTK6 does not indicate that it is a substrate of PTK6. 

BCAR1/p130Cas is a direct substrate

EGFR is a direct substrate.

Correct Ref.

X Li, Y Lu, K Liang, J-M Hsu, C Albarracin, GB Mills, M-C Hung and Z Fan.  Brk/PTK6 sustains activated EGFR signaling through inhibiting EGFR degradation and transactivating EGFR.   Oncogene (2012).

ERK5 correct refs

Ref. Ostrander et al., Cancer Res 2007; 67: (9). May 1, 2007, and Castro et al., 2010 (ref. 73), not ref 63 as indicated.

P38 ref is incorrect for interaction.

Prostate is misspelled as Prostrate in several places.

PTEN interaction was also demonstrated in prostate cells (Wozniak et al., 2017)

Author Response

Dear Editors and Reviewer 1,

We thank Reviewer 1 for taking the time to correct and further improve our manuscript: cancers-2441255, “Therapeutic potential of Protein Tyrosine Kinase 6 in colorectal cancer”. We agree that the accuracy of this review is important, and are grateful for the detail and clarity of their comments. We have addressed them individually below and in the revised manuscript (highlighted in green). We have also re-checked the rest of the manuscript for any typographic errors and consistency of style.

We hope that it will now be of sufficient standard for publication in Cancers and look forward to receiving your decision.

Yours faithfully

Annabelle Lewis, Samanta Jerin and Amanda Harvey

Fig. 1 and Fig. 1 legend. 

  • Labeled residues are not discussed in the figure legend (W44, W184 P175, P177, P179). What is their significance? Need to check this W44 = P175= etc

The following sentence has been added to the legend and referenced as below:

“W44 in the SH3 domain and P175, P177, P179 and W184 in the linker are key for intra-molecular interactions affecting catalytic activity [37]. “

  1. Kim, H.; Jung, J.; Lee, E.S.; Kim, Y.C.; Lee, W.; Lee, S.T. Molecular dissection of the interaction between the SH3 domain and the SH2-Kinase Linker region in PTK6. Biochem Biophys Res Commun 2007, 362, 829-834, doi:10.1016/j.bbrc.2007.08.055.

  • The SH2 domain is not the enzyme catalysis site, the catalytic domain does the work.

The label on the figure has been changed to, “phosphor-tyrosine binding site and protein-protein interaction domain”, to match the description in the main text.

  • Mutation of Y342 does not kill kinase activity but reduces its activation.  The K219 M mutant is kinase-dead.

This has been corrected, with the legend reading as follows:

“Mutation of K219 results in a kinase dead PTK6, mutation of Y342 reduces the activation of PTK6 and mutation of Y447 makes a constitutively active PTK6.”

  • The alternative splice form of PTK6 (ALT-PTK6) is NOT 1354 amino acids long. ALT-PTK6 is only a 15 kDa protein.

This has been corrected to 134aa

  • Lines 77 -79:  The protein description is confusing. N-terminal lipid modification does not occur in the linker. 

This clause has been removed and the information inserted in a separate sentence at the end of the paragraph (line 91).

“Interestingly, unlike the Src family proteins, the N-terminal of PTK6 does not contain a myristoylation signal sequence and is therefore not specifically targeted to the membrane [16].”

  • Lines 87 and 88:  This has not been corrected.  The lysine residue at position 219 (K219) is not phosphorylated.  It is required for ATP binding.  Mutation of K219 to M leads to kinase-dead PTK6.

The sentence has been changed to:

“The tyrosine kinase domain of the PTK6 contains the ATP binding pocket, including the key lysine residue 219, which is required for ATP-binding and phosphotransfer.” 

  • Line 288:  In neonatal mouse highest levels of expression are in the colon, but in the adult mouse highest expression is in the ileum of the small intestine. In both regions, expression is concentrated in more differentiated cells under normal conditions (see Haegebarth et al., 2006, Fig. 2).

This section has now been modified to the following:  “In neonatal mouse, the highest levels of expression are in the colon, but in the adult mouse highest expression is in the ileum of the small intestine. In both regions, expression is concentrated in the more differentiated non-dividing epithelial cells [43,99]. “

  • Lines 229 and 237:  These are redundant.  PTP1b and PTPN1 are the same protein. PTPN1 is the gene name.

The 2nd mention of PTPN1 has been removed.

  • Line 447:  Vemurafenib (PLX4032) was also used to inhibit PTK6 activity in colon cancer cells (Mathur et al., 2016, Fig. 7b).

This information has been added and the paragraph amended as follows:

“Vemurafenib (PLX4032) inhibits PTK6 in both prostate and colon cancer cells, and was able to reduce tumor growth in prostate cancer xenograft models [42,115]. Vemurafenib inhibition of PTK6 presents an interesting avenue for follow up studies in CRC models, since BRAF activating mutations occur frequently in some subtypes of CRC.”

Table 1

  • The heading for column one should read Substrates and Interacting Proteins.  Not all of these proteins are substrates;  co-IP of a protein with PTK6 does not indicate that it is a substrate of PTK6. 

This has been changed

  • BCAR1/p130Cas is a direct substrate

A symbol indicating this has been added

  • EGFR is a direct substrate.

Correct Ref.

X Li, Y Lu, K Liang, J-M Hsu, C Albarracin, GB Mills, M-C Hung and Z Fan.  Brk/PTK6 sustains activated EGFR signaling through inhibiting EGFR degradation and transactivating EGFR.   Oncogene (2012).

A symbol indicating this has been added and the reference changed.

  • ERK5 correct refs

Ref. Ostrander et al., Cancer Res 2007; 67: (9). May 1, 2007, and Castro et al., 2010 (ref. 73), not ref 63 as indicated.

The reference has been corrected

  • P38 ref is incorrect for interaction.

The following reference has been used: Lofgren, K.A.; Ostrander, J.H.; Housa, D.; Hubbard, G.K.; Locatelli, A.; Bliss, R.L.; Schwertfeger, K.L.; Lange, C.A. Mammary gland specific expression of Brk/PTK6 promotes delayed involution and tumor formation associated with activation of p38 MAPK. Breast Cancer Res 2011, 13, R89, doi:10.1186/bcr2946.

  • Prostate is misspelled as Prostrate in several places.

These have been corrected

  • PTEN interaction was also demonstrated in prostate cells (Wozniak et al., 2017)

This has been included and referenced.

Round 4

Reviewer 1 Report

The authors have made the recommended changes and carefully reviewed the final version of the manuscript, it should be ready for publication.